# The Impact of Charlson Comorbidity Index on the Functional Capacity of COVID-19 Survivors: A Prospective Cohort Study with One-Year Follow-Up

**DOI:** 10.3390/ijerph19127473

**Published:** 2022-06-18

**Authors:** Rodrigo Núñez-Cortés, Constanza Malhue-Vidal, Florencia Gath, Gonzalo Valdivia-Lobos, Rodrigo Torres-Castro, Carlos Cruz-Montecinos, Francisco M. Martinez-Arnau, Sofía Pérez-Alenda, Rubén López-Bueno, Joaquín Calatayud

**Affiliations:** 1Physiotherapy in Motion Multispeciality Research Group (PTinMOTION), Department of Physiotherapy, University of Valencia, 46010 Valencia, Spain; r_nunez@uchile.cl (R.N.-C.); carloscruz@uchile.cl (C.C.-M.); francisco.m.martinez@uv.es (F.M.M.-A.); sofia.perez-alenda@uv.es (S.P.-A.); 2Department of Physical Therapy, Faculty of Medicine, University of Chile, Santiago 8380286, Chile; klgorodrigotorres@gmail.com; 3Service of Physical Therapy, Hospital Clínico La Florida, Santiago 8240000, Chile; cpmalhue@uc.cl (C.M.-V.); florencia.gath@redsalud.gov.cl (F.G.); gavaldivia@uc.cl (G.V.-L.); 4Institut d’Investigacions Biomèdiques August Pi i Sunyer (IDIBAPS), 08036 Barcelona, Spain; 5Section of Research, Innovation and Development in Kinesiology, Kinesiology Unit, San José Hospital, Santiago 8380286, Chile; 6Department of Physical Medicine and Nursing, University of Zaragoza, 50009 Zaragoza, Spain; 7National Research Centre for the Working Environment, 2100 Copenhagen, Denmark; joaquin.calatayud@uv.es; 8Exercise Intervention for Health Research Group (EXINH-RG), Department of Physiotherapy, University of Valencia, 46010 Valencia, Spain

**Keywords:** coronavirus disease 2019, physical capacity, six-minute walk test, comorbidity

## Abstract

Objective: To determine the association between the Charlson comorbidity index (CCI) score after discharge with 6-min walk test (6MWT) 1 year after discharge in a cohort of COVID-19 survivors. Methods: In this prospective study, data were collected from a consecutive sample of patients hospitalized for COVID-19. The CCI score was calculated from the comorbidity data. The main outcome was the distance walked in the 6MWT at 1 year after discharge. Associations between CCI and meters covered in the 6MWT were assessed through crude and adjusted linear regressions. The model was adjusted for possible confounding factors (sex, days of hospitalization, and basal physical capacity through sit-to-stand test one month after discharge). Results: A total of 41 patients were included (mean age 58.8 ± 12.7 years, 20/21 men/women). A significant association was observed between CCI and 6MWT (meters): (i) crude model: β = −18.7, 95% CI = −34.7 to −2.6, *p* < 0.05; (ii) model adjusted for propensity score including sex, days of hospitalization, and sit-to-stand: β = −23.0, 95% CI = −39.1 to −6.8, *p* < 0.05. Conclusions: A higher CCI score after discharge indicates worse performance on the 6MWT at 1-year follow-up in COVID-19 survivors. The CCI score could also be used as a screening tool to make important clinical decisions.

## 1. Introduction

Although complete vaccination against COVID-19 has covered nearly 60% of the world’s population, the pandemic continues to grow, and cumulative cases have exceeded 520 million worldwide, with more than 6 million deaths reported [1]. In those who survive, a wide variety of sequelae have been described [2]. Among others, survivors may experience fatigue, persistent respiratory symptoms, decreased physical function, and decreased quality of life for up to 6 months post-infection [3].

In general, individuals with comorbidities are more affected by COVID-19 and have worse clinical outcomes [4]. However, due to the high prevalence of impaired skeletal muscle strength and physical performance in patients without previous disabilities recovering from COVID-19 [5], it remains a challenge for rehabilitators to objectify and stratify changes in physical function for future decision making. A valid, reliable, and simple tool for risk stratifying patients based on comorbidities is the Charlson comorbidity index (CCI) [6], which has been widely used as a predictor of prognosis and death in patients with COVID-19 [7,8]. A recent meta-analysis showed that an increase in CCI score was prognostically associated with mortality and poor outcomes in patients hospitalized for COVID-19 [9]. In particular, CCI score ≥ 3 is associated with increased mortality and disease severity in patients with COVID-19 [9], albeit it is unknown whether this risk factor could be useful in determining physical function and quality of life one year after discharge.

To evaluate functional capacity in patients with chronic respiratory disease, the 6-min walk test (6MWT) is the reference validated test [10]. Previous studies found that distance walked on the 6MWT is independently associated with all-cause mortality in older adults and patients with chronic respiratory disease [11,12]. However, the 6MWT needs time, close supervision, and specific technical requirements (e.g., a 30-m corridor) [10]. Additionally, during the first weeks after discharge most of the COVID-19 survivors are unable to complete the 6MWT [13]. Therefore, other clinical tests have been alternatively considered to measure the functional consequences of hospitalization after COVID-19, such as the one-minute sit-to-stand (1-STS) test, the Short Physical Performance Battery (SPPB) or the Timed Up and Go test (TUG) [14,15,16].

Function and mortality are related, and comorbidity burden may be used to predict both outcomes [17]. In this context, the CCI score might be used as a screening tool to make important clinical decisions, such as the allocation of scarce healthcare resources for the rehabilitation of COVID-19 survivors. Importantly, this questionnaire can be administered to several patients at a time (which can be especially relevant during a pandemic and when there are limited resources) and could be used in frail patients with reduced physical capacity to perform demanding physical tests [18,19]. This could accelerate decision-making to start with the recovery process, guiding rehabilitators to provide more specialized interventions in COVID-19 survivors. The main objective of this study was to determine the association between the CCI score after discharge with 6MWT 1 year after discharge in a cohort of COVID-19 survivors. The secondary objective was to assess changes over time in functional capacity using the 1-STS.

## 2. Materials and Methods

### 2.1. Study Setting, Design, and Participants

A prospective cohort study was conducted in a consecutive sample of patients with laboratory-confirmed COVID-19 infection discharged from Hospital Clínico la Florida (Santiago, Chile) between 4 August 2020 and 11 September 2020. Inclusion criteria were as follows: (I) patients over 18 years of age; (II) diagnosis of COVID-19 by reverse transcription-polymerase chain reaction (PCR); and (III) outpatients discharged from the hospital one month prior to study entry. Participants with severe mobility reduction or hemodynamic instability (systolic blood pressure > 180 mmHg or diastolic blood pressure > 100 mmHg) were excluded. The study was conducted according to the guidelines of the Declaration of Helsinki and approved by the Institutional Review Board of the Clinical Hospital la Florida (Santiago, Chile, HLF07102020RNC). Written informed consent was obtained from all participants during enrolment. This study was performed according to the guidelines “Strengthening the Reporting of Observational Studies in Epidemiology” (STROBE) [20].

### 2.2. Data Collection (One Month after Discharge)

One month after discharge, patients were admitted to the follow-up program at the hospital. At admission, demographic characteristics (age, sex, and body mass index), days of hospitalization, and underlying comorbidities were collected. The functional capacity was measured by the number of repetitions the subjects were able to sit and stand during the 1-STS; tests were performed with the same standard height chair (46 cm) without armrests placed against the wall [21]. Subjects could not use their hands/arms to push the chair seat or their body. Patients were instructed to complete as many sitting and standing cycles as possible in 1 min at a self-paced pace. Reference values based on the healthy adult population previously reported by Strassmann et al. were used [22].

### 2.3. Charlson Comorbidity Index (One Month after Discharge)

The CCI score was calculated from the following information: age-based scoring starts at ≥50 years, with an increase of one point for every 10 years; history of definite or probable myocardial infarction (+1 point); congestive heart failure (+1 point); peripheral vascular disease (+1 point); cerebrovascular disease (+1 point); dementia (+1 point), chronic obstructive pulmonary disease (COPD) (+1 point); connective tissue disease (+1 point); peptic ulcer disease (+1 point); liver disease (mild, +1 point; moderate to severe, +3 point); diabetes mellitus (+1 point); hemiplegia (+2 point); moderate to severe chronic kidney disease (+2 point); solid tumor (localized, +2 point; metastasis, +6 point); leukemia (+2 point); malignant lymphoma (+2 point); and acquired immune deficiency syndrome (+6 point) [6]. Data were collected by one researcher and then verified by a second researcher.

### 2.4. Follow-Up

All patients were admitted to a home rehabilitation program, which included 7–10 therapeutic exercise sessions over 3–4 weeks. Each session lasted approximately 30 min and included breathing exercises, walking, and functional exercises of the upper and lower extremities, depending on the patients’ need and tolerance. After completing the rehabilitation program, participants received printed material with recommendations for continuing with general mobility exercises (gluteal isometrics, squats with upper limb support, and static walking). One year after discharge, patients returned to the hospital outpatient unit to assess functional capacity using the 6MWT. In addition, the 1-STS was re-evaluated under the same conditions as the first evaluation.

### 2.5. 6-Minute Walk Test (1 Year after Discharge)

The 6MWT test was conducted following international recommendations [10]. Participants were instructed to walk as far as possible in a 30-m indoor corridor, pausing to rest if needed. The modified Borg scale (0–10) was used to measure dyspnea and fatigue immediately before and after the test. SpO_2_ and HR were continuously monitored during the test. To estimate the predicted 6MWT distance we used the reference values for 6MWT in the Chilean adult population [23]. The formula for estimating the distance traveled was: (i) women: 457 − 3.46 × age (years) + 2.61 × height (cm) − 1.57 × weight (kg) ± 53; (ii) men: 530 − 3.31 × age (years) + 2.36 × height (cm) − 1.49 × weight (kg) ± 58 [23].

### 2.6. Statistical Analyses

We conducted statistical analyses through Stata version 16.1 (StataCorp, Texas, TX, USA). The Shapiro test served to check normality. Associations between CCI (exhibition) and meters covered in the 6MWT (outcome) were assessed through crude and adjusted linear regressions. Due to the low number of individuals, we created a propensity score index that accounted for potential confounders, such as sex, days of hospitalization due to COVID-19, and previous functional capacity (i.e., 1-STS one-month after discharge), for model adjustment. We checked the final model using additional variables without improving the accuracy of the obtained estimations. Estimations are provided as beta coefficients and 95% confidence intervals (CI). Finally, we created a scatterplot showing the data distribution of the two examined variables with a regression line and a shadowed 95% CI. Differences over time for 1-STS were evaluated using the Wilcoxon test or paired *t*-test according to the distribution of the data. Levels of significance were set at *p* < 0.05. Post hoc power (1-β err prob) was calculated with the G*Power 3.1 program, version 3.1.9.2 (Universität Düsseldorf, Düsseldorf, Germany), using the linear multiple regression statistical test (single regression coefficient). Input data were: one-tailed; effect size f^2^ = 0.59; α err prob = 0.05; total sample = 41; and number of predictors = 1.

## 3. Results

A total of 45 patients met the selection criteria and were evaluated one month after discharge. Four patients were lost to 1-year follow-up (Figure 1). Participants with missing data in any study variable were discarded for the study. Finally, a sample of 41 COVID-19 survivors was analyzed. The median age was 58.8 ± 12.7 years, and 20 (48.8%) patients were male. Seventy percent of the patients had values below the estimated 6MWT value. The characteristics of COVID-19 survivors at hospital discharge and at 1-year follow-up are shown in Table 1.

A significant association was observed between CCI and distance covered (meters) in the 6MWT one year after hospitalization for COVID-19: (i) crude model: R^2^ = 0.1022, β = −18.7, 95% CI = −34.7 to −2.6, *p* < 0.05; (ii) adjusted model: R^2^ = 0.3286, β = −23.0, 95% CI = −39.1 to −6.8, *p* < 0.05 (Appendix A). Figure 2 shows the scatter plot with the regression line for the crude association between CCI and 6MWT. Fifty-two percent of COVID-19 survivors had a CCI score ≥ 3 (high comorbidity burden). The post hoc power (1-β err prob) of association between the CCI and the 6MWT was 99%. Finally, a significant increase in the 1-STS of 3.5 repetitions was observed during follow-up (*p* < 0.05). However, 1-year after discharge, 85% of participants were below the 25th percentile, relative to baseline values [22].

## 4. Discussion

The results of this study showed that there is a significant association between CCI and distance on the 6MWT in COVID-19 survivors 1 year after discharge. Specifically, a higher CCI score one month after discharge was an indicator of worse performance on the 6MWT at 1-year follow-up. Although we observed a small improvement in 1-STS over time, 70% of patients had values below the estimated 6MWT value at one year after discharge. Given that most patients are unable to perform the 6MWT at discharge or one month after discharge, the 1-STS appears to be a reliable functional test that can provide a parameter to monitor functional status over time. However, considering that the 6MWT is the gold standard for assessing functional capacity, it may provide more relevant information when patients are already able to perform the test [10]. Thus, the CCI score is a good estimate of long-term physical function. Our linear regression model assumes that a one-point increase in the CCI score reduces 23 m in the 6MWT. To date, this is the first study investigating an affordable tool to assess long-term physical recovery among previously hospitalized COVID-19 patients.

These findings are not unexpected, as comorbidity is a clinical condition of exposure that influences different health outcomes [24]. Although CCI was initially developed to predict mortality, it has been shown to predict functional outcomes in other populations [19]. For example, the CCI score was independently associated with functional outcome and mortality six months after a cerebrovascular event (i.e., ischemic stroke and intracerebral hemorrhage) [25]. Even in stroke patients, it appears to be better than pathology-specific comorbidity indexes [26]. In contrast, Groll et al. [27] reported that CCI would not be related to long-term physical function in acute respiratory distress syndrome patients. However, physical function was not objectively measured in their study since they used the SF-36 physical function subscale [27].

Although the use of the 6MWT in COVID-19 survivors is scarce, the distance walked in the 6MWT is an independent risk factor influencing activities of daily living and all-cause mortality in patients with chronic respiratory disease and in older adults [11,12,28]. Thus, these findings support evidence suggesting that function and mortality are related, and that comorbidity burden can be used to predict both outcomes [17].

Our results suggest that physical function 1 year after discharge from COVID-19 could be estimated using the CCI score. Our linear regression model assumes that a one-point increase in the CCI score reduces 23 m in the 6MWT. Since the R^2^ of the linear regression was small (R^2^ = 0.33), there might be other factors not considered by our study that influence 6MWT at 1-year follow-up after COVID-19. However, the CCI proved to be an important parameter to guide clinical management at discharge. Given the large number of patients with sequelae after COVID-19, the CCI could be used to discriminate against patients who may develop long-term limitations in their physical function. In addition, the CCI could be added to follow-up programs, as it could help discriminate those patients who could benefit from interventions, such as physical rehabilitation or at least closer follow-up. Multidisciplinary rehabilitation teams should also be aware of the impact of social determinants of health on the severity of COVID-19 to promote strategies that improve the long-term functionality of these patients [29].

Our results indicate that patients with a higher comorbidity burden had poorer performance on the 6MWT. These results could be partly explained by the association between comorbidity burden and sarcopenia [30,31]. In particular, Gong et al. [32] found a strong and significant negative correlation between CCI and skeletal muscle index/gait speed in older adults. In fact, indirect evidence of the relationship between skeletal muscle function and the 6MWT could be the effectiveness of resistance training to improve this test in patients with COPD [33]. In addition, a recent meta-analysis found that pre-existing comorbidity, together with female sex and respiratory distress syndrome, were associated with reduced 6MWT results after critical illness [34].

The CCI score may be especially useful when there are time constraints and limited healthcare resources or when the patient cannot perform demanding physical tests, such as the 6MWT, after hospitalization for COVID-19. For example, the CCI could be prioritized among frail patients with limited balance or muscle strength at increased risk of falling or patients with hemodynamic instability (e.g., a systolic blood pressure > 180 mmHg) [10]. On the other hand, considering that the incidence of comorbidity increases significantly with age in older adults [35] and that disability in this group is much higher when two or more diseases coexist [36], the CCI may be used in the early phases of rehabilitation planning after hospitalization for COVID-19.

The strengths of our study comprise the use of a monitored sample of confirmed COVID-19 survivors after hospitalization with 1-year follow-up. We also used validated tools to assess exposition, outcome, and covariates. However, the findings of this study should be considered in light of several limitations. First, the sample size is small, which may lead to a certain degree of selection bias, but still enough to detect relevant changes in the examined variables, and the findings are consistent. To overcome this limitation, the post hoc power of the study was calculated and was high (99%). Second, there is still the possibility of a residual confounding bias (e.g., due to lack of adjustment for physical activity variables), although the chance that this can significantly vary the findings is low due to the use of relevant potential confounders (e.g., 1-STS). Finally, due to the specific context where the study was conducted, generalizations over other populations should be cautiously taken. Future studies should also assess whether there is a prospective association between other measures that help to predict the recovery in COVID-19 survivors (e.g., skeletal muscle mass evaluation).

## 5. Conclusions

A higher CCI score one month after discharge indicates worse performance on the 6MWT at 1-year follow-up in COVID-19 survivors. The CCI score could also be used as a screening tool to make important clinical decisions, such as the allocation of scarce healthcare resources or to accelerate and personalize rehabilitation.

## Figures and Tables

**Figure 1 ijerph-19-07473-f001:**
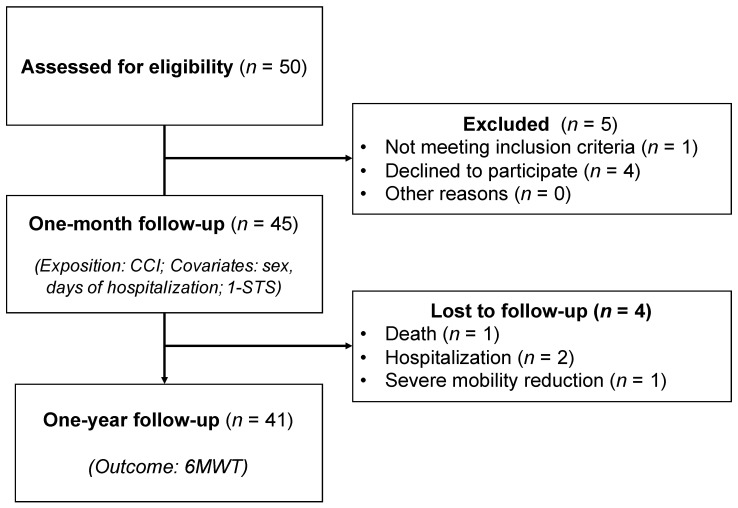
Patient flow diagram. **Abbreviations**: 1-STST: 1-min sit-to-stand; 6MWT, 6-min walk test; CCI: Charlson comorbidity index.

**Figure 2 ijerph-19-07473-f002:**
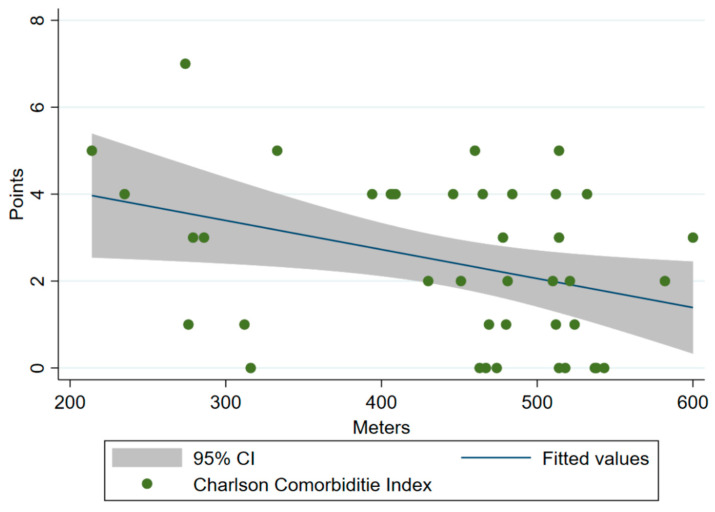
Scatter plot with regression line showing crude association between Charlson Comorbidity Index and distance covered (meters) in the 6MWT test one year after COVID-19 hospitalization.

**Table 1 ijerph-19-07473-t001:** Patient characteristics.

Characteristic	Total (*n* = 41)
Age (mean ± SD)	58.8 ± 12.7
Sex, male, *n* (%)	20 (48.8)
BMI (mean ± SD)	30.7 ± 5.2
Invasive ventilation, *n* (%)	12 (28.3)
Hospitalization days (mean ± SD)	13.5 (4 to 55)
1-STS (repetitions)	20 (12 to 29)
**Comorbidities, *n* (%)**	
Myocardial infarction	2 (4.9)
Congestive heart failure	2 (4.9)
Peripheral vascular disease	24 (58.5)
Cerebrovascular accident	1 (2.4)
COPD	4 (9.8)
Liver disease	1 (2.4)
Diabetes mellitus	14 (34.1)
Chronic kidney disease	2 (4.9)
Cancer	2 (4.9)
Charlson comorbidity index (points)	2 (0 to 8)
**One-year follow-up**	
1-STS (repetitions)	23.5 (12 to 35)
6MWT (meters)	443.5 (214 to 600)
6MWT (%predicted)	80.5 ± 15

**Abbreviations:** 1-STST: 1-min sit-to-stand; 6MWT, 6-min walk test; BMI: body mass index; COPD: chronic obstructive pulmonary disease. Values are mean ± standard deviation, median (range) or percentages.

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
