# Peer review of "The Impact of Charlson Comorbidity Index on the Functional Capacity of COVID-19 Survivors: A Prospective Cohort Study with One-Year Follow-Up"

_ijerph, 2022, doi:10.3390/ijerph19127473_

Round 1
Reviewer 1 Report
The paper is methodologically correct, the results are well represented, it's possible to add the table of crude and adjusted linear regressions also as supplementary material.
Author Response
We thank Reviewer 1 for his comment, we have added the table of raw and fitted linear regressions also as supplementary material.We hope that the revised version of our manuscript has met reviewers expectations.

Reviewer 2 Report
Summary
The authors present a small prospective cohort study of 41 patients which had been hospitalized for COVID-19, and followed for 1 year. The authors use a robust analysis method to account for the small sample size, mainly adjusting for a propensity score rather than individual covariates. Overall, the findings of the study appear to be sound and suggest that the CCI may be a predictor of long-term physical impairments in people who have had COVID-19.
Abstract
Please define CCI prior to first use in the abstract.
The authors mention that they undertook adjustment in the methods, but then mention that they actually adjusted for the propensity score in the results section. Please change the statement in the methods section to reflect that you were adjusting for a propensity score that including those variables as this is not the same thing as adjusting directly for these variables.
Methods
The authors state “we created a propensity score index that accounted for potential confounders such as sex, days of hospitalization due to COVID-19, and previous functional capacity (i.e. 1-STS one-month after discharge) for model adjustment”. Is this the totality of variables adjusted for in the propensity score or were there others? One would expect age and BMI to also be adjusted for.
Can the authors also please clarify what model they used to propose the propensity score (including the log and link function)? At the moment, I am not clear what you were modeling based on what is presented.
Can the authors please clarify when they did the respective tests as well? If the patients were evaluated one month after discharge, was the 6 minute walk test taken then or at another follow-up 1 year later. What exactly was taken at the 1 month follow-up? If the sit to stand was taken. I am also unclear when the CCI was taken. Was this prior to or after the COVID-19 infection?
Results
The authors keep mentioning this 3 or greater cut off for the CCI yet do not mention a statistical test they would have used to test for this association. Please update this in the statistical analysis section. Also, please also present the propensity scored adjusted association.
The authors state “The power of association between the CCI and the 6MWT was 99%”. I am not sure what this means. Please elaborate.
Discussion
The authors state “the post hoc power of the study was calculated and was high (99%)”. Can they please provide more information regarding which test they ran exactly in g*power and what input values they used?
Author Response
We appreciate the comments that contributed to improving the quality of the manuscript significantly. We hope that the revised version of our manuscript has met reviewers expectations.
1) Please define CCI prior to first use in the abstract.
R: Thank you, we have made the change you requested.
2) The authors mention that they undertook adjustment in the methods, but then mention that they actually adjusted for the propensity score in the results section. Please change the statement in the methods section to reflect that you were adjusting for a propensity score that including those variables as this is not the same thing as adjusting directly for these variables.
R: We appreciate your comments. The following sentence is included in the Methods section: “Due to the low number of individuals, we created a propensity score index that accounted for potential confounders such as sex, days of hospitalization due to COVID-19, and previous functional capacity (i.e. 1-STS one-month after discharge) for model adjustment.”
3) The authors state “we created a propensity score index that accounted for potential confounders such as sex, days of hospitalization due to COVID-19, and previous functional capacity (i.e. 1-STS one-month after discharge) for model adjustment”. Is this the totality of variables adjusted for in the propensity score or were there others? One would expect age and BMI to also be adjusted for.
R: We thank you for raising this point. Because the Charlson comorbidity index includes age as one of the score points, we avoid overadjustment in the analyses. Similarly, because BMI and other variables did not change the results of the created model, we adhered to the parsimonious principle to have the more fitted model with the lower possible variables. We have now underscored this in the Methods section: “We checked the final model using additional variables without improving accuracy of the obtained estimations.”
4) Can the authors also please clarify what model they used to propose the propensity score (including the log and link function)? At the moment, I am not clear what you were modeling based on what is presented.
R: We thank the reviewer for his comment. To clarify this point, we have specified that we performed a linear regression with the outcome (6MWT meters), the exposure (CCU), and adjusted for the propensity score matching we created with the covariates, report the beta coefficients obtained
5) Can the authors please clarify when they did the respective tests as well? If the patients were evaluated one month after discharge, was the 6 minute walk test taken then or at another follow-up 1 year later. What exactly was taken at the 1 month follow-up? If the sit to stand was taken. I am also unclear when the CCI was taken. Was this prior to or after the COVID-19 infection?
R: We thank Reviewer 2 for this important comment. We are sorry for not being clear with this information, we have provided more details on when they did the respective tests in the sub-headings of the methods section. Furthermore, in figure 1 we specify when each variable was assessed: i) one month after discharge: STS, CCI, covariates; ii) one month after discharge: 6MWT, STS.
6) The authors keep mentioning this 3 or greater cut off for the CCI yet do not mention a statistical test they would have used to test for this association. Please update this in the statistical analysis section. Also, please also present the propensity scored adjusted association.
R: Thank you for your comment. To address your concern, we have made the following changes: 1) We have removed the analysis related to the cutoff point of 3 or more for the CCI, as this was not the primary objective of the study and may present confusion; 2) As commented, we have now detailed information on this point in the Statistics subsection. “Due to the low number of individuals, we created a propensity score index that accounted for potential confounders such as sex, days of hospitalization due to COVID-19, and previous functional capacity (i.e. 1-STS one-month after discharge) for model adjustment.”
7) The authors state “The power of association between the CCI and the 6MWT was 99%”. I am not sure what this means. Please elaborate.
R: Thank you for this comment, we have specified that it refers to post hoc power (1-β err prob).
8) The authors state “the post hoc power of the study was calculated and was high (99%)”. Can they please provide more information regarding which test they ran exactly in g*power and what input values they used?
R: Thank you for your comments, we have now added the following information: “Post hoc power was calculated with G*Power 3.1, using the linear multiple regression statistical test (single regression coefficient). Input data were: one-tailed, effect size f2: 0.59, α err prob: 0.05, total sample 41, number of predictors: 1.”

Reviewer 3 Report
Dear Editor, I appreciate the opportunity to review this interesting study. Overall, the authors address the association between Charlson comorbidity index, assessed 1 month after the hospital discharge post COVID-19, and Functional Capacity 1 year after the hospital discharge. The rationale behind is that probably those who worse CCI index, would present worst functional capacity in the 1-year follow-up. This topic is interesting for patients and clinicians dealing with COVID-19. However, some points needs to be addressed and improved in the manuscript. Please, find below my comments and suggestions:
1) Quality of life is mentioned in the introduction and objective of the study. However, this is not addressed in the remaining of the manuscript (i.e. methods and results).
2) Sit-to-Stand Test is not mentioned in the objective of the study but is one of the main outcome. I suggest authors reformulate their objective by using "functional capacity" instead of only 6MWT.
METHODS
3) Page 2, Line 88: Provide the number of the protocol approval by the Ethical Committee.
4) To better describe the sample, underlying comorbidities could be presented at Table 1.
5) What were the instructions to the patients after the 4-week rehab program? Did they keep with some regular exercises? Did they receive instructions to have some "good" habits regarding exercise, diet etc. or they were instructed to keep with their normal routine?
6) Also, it is important to have more information on the patients at the follow-up. For example, have you assessed the level of physical activity at 1 year after discharge? Have the patients been hospitalized or presented any other complication during the 1-year follow-up? This information should be presented.
7) Provide in the methods section the formula and its reference used to estimate the predicted 6MWT distance.
RESULTS
8) Provide the adjusted R2 for the regression analyses.
9) It was not in the study's objective to compare groups according to the CCI or determine CCI cutpoint (<3). It could be inserted at the objective and authors must provide the statistical analysis used to compare the groups.
10) Page 5, Line 161: It is not clear what the authors mean by the "power of association". Is that the adjusted R2? Clarify.
11) Page 5, Lines 175/176: It is not clear how 6MWT is better than 1-STS based on the argument used by the authors. I assume it was expected that after 1 year of discharge patients would be better than after 1 month after the discharge. Actually, it appears to be the opposite (1-STS might be better than the 6MWT). Since most patients cannot perform 6MWT at discharge or 1 month after the discharge, the 1-STS seems to be a more reliable functional test and can provide a parameter for monitoring the functional status overtime. Please, comment on this.
12) Have you performed a linear regression between 1-STS and CCI?
Author Response
1) Quality of life is mentioned in the introduction and objective of the study. However, this is not addressed in the remaining of the manuscript (i.e. methods and results).
R: We thank the reviewer for this important comment. We regret this error, quality of life was not a variable assessed by us, therefore, we have removed it from the objective of the study.2) Sit-to-Stand Test is not mentioned in the objective of the study but is one of the main outcome. I suggest authors reformulate their objective by using "functional capacity" instead of only 6MWT.
R: Since the primary objective of the study is in line with the gold standard of outcomes for assessing functional capacity (i.e. 6MWT), we have mentioned STS as a secondary objective.3) Page 2, Line 88: Provide the number of the protocol approval by the Ethical Committee.
R: We thank the Reviewer 3 for this comment, we have incorporated the requested information.
4) To better describe the sample, underlying comorbidities could be presented at Table 1.
R: Thank you, we have incorporated the underlying comorbidities for the total sample in Table 1.
5) What were the instructions to the patients after the 4-week rehab program? Did they keep with some regular exercises? Did they receive instructions to have some "good" habits regarding exercise, diet etc. or they were instructed to keep with their normal routine?
R: Thank you for your comments, we have now added the following information: “After completing the rehabilitation program, participants received printed material with recommendations for continuing with general mobility exercises (gluteal isometrics, squats with upper limb support, static walking)”.6) Also, it is important to have more information on the patients at the follow-up. For example, have you assessed the level of physical activity at 1 year after discharge? Have the patients been hospitalized or presented any other complication during the 1-year follow-up? This information should be presented.
R: We thank the reviewer for this comment. Although we did not consider the level of physical activity, we have used an outcome related to physical capacity and a covariate also related to physical (1-STS), both objectively measured. It is possible that this is controlled for in some way, and that using physical activity does not add anything new to the model. However, we have specified the following in the study limitations:
“Second, there is still the possibility of a residual confounding bias (e.g., due to lack of adjustment for physical activity variables), although the chance that this can significantly vary the findings is low due to the use of relevant potential confounders (e.g., 1-STS)“
7) Provide in the methods section the formula and its reference used to estimate the predicted 6MWT distance.
R: Thank you for your comment, we have specified this information under the sub-heading "6-minute walk test" in the methods section.
“To estimate the predicted 6MWT distance we used the reference values for 6MWT in the Chilean adult population.” The formula for estimating the distance traveled was: i) Women: 457-3.46 x age (years) + 2.61 x height (cm) -1.57 x weight (kg) ± 53; ii) Men: 530-3.31 x age (years) + 2.36 x height (cm) -1.49 x weight (kg) ± 58 [23].
https://pubmed.ncbi.nlm.nih.gov/21249280/
8) Provide the adjusted R2 for the regression analyses.
R: We thank reviewer 3 for this important comment; following his recommendations, we have provided the adjusted R-squared for the regression analyses in the results section. In addition, we have added the table of raw and fitted linear regressions also as supplementary material.9) It was not in the study's objective to compare groups according to the CCI or determine CCI cutpoint (<3). It could be inserted at the objective and authors must provide the statistical analysis used to compare the groups.
R: Thank you for your comment, we have removed the analysis related to the cutoff point of 3 or more for the CCI, as this was not the primary objective of the study and may present confusion. Thus, we present the results without considering the CCI score as a dichotomous variable.10) Page 5, Line 161: It is not clear what the authors mean by the "power of association". Is that the adjusted R2? Clarify.
R: Thank you for this comment, we have specified that it refers to post hoc power (1-β err prob).
11) Page 5, Lines 175/176: It is not clear how 6MWT is better than 1-STS based on the argument used by the authors. I assume it was expected that after 1 year of discharge patients would be better than after 1 month after the discharge. Actually, it appears to be the opposite (1-STS might be better than the 6MWT). Since most patients cannot perform 6MWT at discharge or 1 month after the discharge, the 1-STS seems to be a more reliable functional test and can provide a parameter for monitoring the functional status overtime. Please, comment on this.
R: We thank the reviewer for this important observation. To avoid a value judgment, we have removed the sentence indicating, moreover, it was not our objective which of the two tests was better. On the other hand, we agree with what reviewer 3 mentioned, so we have specified in the discussion the following: "Given that most patients cannot perform the 6MWT at discharge or 1 month after discharge, the 1-STS seems to be a reliable functional test that can provide a parameter to monitor functional status over time. However, considering that the 6MWT is the gold standard for assessing functional capacity, it could provide relevant information when patients are already able to perform the test."12) Have you performed a linear regression between 1-STS and CCI?
R: Considering that 6MWT is the gold standard for assessing functional capacity, we used this variable as the primary outcome. Therefore, it was not part of our objective to assess the association with 1-STS.

Round 2
Reviewer 3 Report
Dear editor, I thank you for the opportunity to review such interesting study. All the points I raised were properly addressed by the authors. I have only one minor comment:
Since the R2 of the linear regression is quite low (R2 = 0.33), authors must acknowledge in their discussion that there are other factors not considered by the study that might influence on the 6MWT at 1-year follow-up after COVID-19. Even though, CCI proved to be an important parameter to guide the clinical management at discharge.
Kind regards
Author Response
We thank the reviewer for this valuable comment. Following his suggestion, we have added the following sentence to the fourth paragraph of the discussion:
“Since the R2 of the linear regression was small (R2=0.33), there might be other factors not considered by our study that influence 6MWT at 1-year follow-up after COVID-19. However, the CCI proved to be an important parameter to guide clinical management at discharge”.
